# 3D Printed (Binder Jetting) Furan Molding and Core Sands—Thermal Deformation, Mechanical and Technological Properties

**DOI:** 10.3390/ma16093339

**Published:** 2023-04-24

**Authors:** Artur Bobrowski, Karolina Kaczmarska, Dariusz Drożyński, Faustyna Woźniak, Michał Dereń, Beata Grabowska, Sylwia Żymankowska-Kumon, Michał Szucki

**Affiliations:** 1Faculty of Foundry Engineering, AGH University of Krakow, Reymonta 23, 30059 Krakow, Poland; 2Foundry Institute, TU Bergakademie Freiberg, 09599 Freiberg, Germany

**Keywords:** 3D printing, molding sand, core sand, furfuryl resin, thermal deformation

## Abstract

Casting cores produced in additive manufacturing are more often used in industrial practice, in particular in the case of the production of unit castings and castings with very complex geometry. The growing interest in the technology of 3D printing of cores and molds also brings emerging doubts related to their mechanical and technological properties. This article presents a comparison of the properties of cores made of sand with acid-curing furfuryl resin, made with 3D printing technology; the cores were prepared in a conventional way (mixing and compaction). The main purpose of this research was to determine the possibility of using shell cores as a substitute for solid cores, aimed at reducing the amount of binder in the core. The influence of the type of the binder and the size of the grain matrix fraction on the obtained mechanical and technological properties of the cores, with particular emphasis on abrasion and thermal deformation, as well as on the kinetics of their hardening, was demonstrated.

## 1. Introduction

Foundry is one of the oldest methods of manufacturing usable parts. Over the years, it has remained one of the basic methods of producing parts for the engineering, automotive, aviation, shipbuilding, energy, and many other industries. The main advantage of casting is the possibility of making elements of complex shape with high mechanical properties that are difficult or even impossible to obtain through other technological processes. The process involves melting and then pouring a previously prepared mold, which imitates the future shape of the casting, with a liquid casting alloy of a specific composition. Most castings are made in disposable molds using molding and core sands, which, by definition, are a mixture of quartz sand, binding material, and additives. There are many bonding systems, of which molding sands with bentonite are the most commonly used due to their low cost and the possibility of using them in a closed circuit. However, their limitation is the low quality of the surface. In the production of high-quality castings, organic binders based on furfuryl resin are a frequently used system for binding grain matrix grains. They meet the basic requirements for molding and core sands, i.e., [1,2]:Durability and time-regulated shelf life for molding;High resistance-to-moisture during core storage;High resistance-to-erosion and -penetration and no chemical interaction with the casting alloy;High gas permeability;High strength;Easy knockout properties;Good susceptibility to regeneration of used molding sand;Lower capital expenditure;No need to heat core boxes;High surface quality of castings;Environmentally friendly core systems with low odor.

The problem of gas and the emission of odors is widely discussed in publications [3,4,5,6]. In this regard, molding and core sand systems based on organic binders should not be considered the most environmentally friendly due to the emission of BTEX group compounds, which is an indicator of the environmental harm of the molding sands. Therefore, in recent years, much attention has been paid to inorganic molding and core sand systems [7,8,9,10,11,12]. Their significant advantage is their low emissivity, but they are not without disadvantages, including poor knockout properties [13,14], a limited reclamation ability, and unfavorable effects on bentonite (deactivation). Resin-coated sands are a widely used alternative that guarantees a very good surface with minimal defects in gray and ductile iron castings, according to [15,16,17]. Due to their good mechanical and technological properties, as well as the possibility of obtaining cores and complex shape geometries, but also their susceptibility to regeneration, it is expedient to use cores made with binders of organic-type binders [18,19,20,21,22]. The high mechanical properties of the molds and cores allow planners and designers to save more materials, not only in terms of consumption of liquid casting alloy but also in terms of materials for making the mold, which allows for significant savings in the production process. In the last ten years, tremendous progress has been made in the field of 3D printing of molds and sand cores using binder-jetting technology.

The manufacture of molds and cores using 3D sand printing is of great interest, as evidenced by the growing number of publications devoted to the subject [23,24,25,26]. One of the most important advantages is the virtually unlimited possibilities regarding the shape of cores, as well as the high precision of printing. Printer manufacturers offer various types of a wide range of bonding systems, which are either being improved, or new ones are produced each year. Among them, organic furan and phenolic binders are the most popular [27], but in recent years, inorganic binders have received the most attention [23,28,29,30].

The process of printing molds and cores differs significantly from the conventional method of making molding sands with acid-cured furfuryl resin. First of all, in the conventional method, mixing is used, which allows for a fairly uniform coating of the binder over the entire surface of the matrix grains. In the case of 3D printing, we are dealing with the imposition of a portion of sand, mixed beforehand with a suitable amount of hardener, on a previously applied thin layer of resin. There is no additional compaction process here, leading to a lower packing density. This can promote free spaces between the grains, and the printed parts may have increased porosity. The manufacturer of the ExOne (Huntingdon, PA, USA) [31] printer, therefore, recommends using a fine grain matrix. However, it should be borne in mind that when applying a layer of binder to a porous bed such as the previous layer of the built core, pore filling may occur. Consequently, there may be less binder on the surface of the grains, as it has been used up to fill the pores between the sand grains. More binder may need to be added to achieve adequate strength. Such conclusions were reached by the authors of [24,27], who clearly indicate that the reason for the inferior surface quality of castings for the production of which 3D-printed cores were used is directly linked to excess binder. The authors of [1,32,33,34], on the other hand, demonstrated the significant role of correct printing parameter settings, such as printing speed or the pressure of the newly applied sand layer, in reducing core porosity, which plays a decisive role in shaping a technologically important parameter—permeability. This property determines the ability of molding sand to carry away the gaseous products of binder decomposition. The authors point to a compromise between adequate permeability to reduce gas-related defects and the susceptibility of the core to the penetration of liquid metal between grains, which affects the surface-casting surface quality. The authors of [35] initially assumed that printed cores (ExOne printer) would exhibit similar mechanical properties to a typical binder (trade name: Super Set 942 resin, TW30 hardener). Using the molding sand composition, one part by weight of resin-to-sand and 35% of hardener-to-resin, they obtained more than three times higher tensile strength for 3D-printed specimens. They concluded that the obtained better mechanical performance for the printed cores could be due to the even and fully controlled process of applying layers of sand and resin. They further showed that the cores obtained on the ExOne printer had greater stability at an elevated temperature. However, the authors’ statement that the printed samples contained a low binder content for gas generation is questionable, as for the Super Set system the roasting loss is about 1.2%, while for 3D printing it is higher (1.4%). Hence, the expected amount of binder decomposition products should be higher.

Three-dimensionally printed cores using furfuryl resin have an indisputable advantage over the conventional self-cured molding sand system due to the slower heat dissipation process and smaller pores between grains while printing the core in a bed of loose sand. This advantage is due to the slower heat dissipation process and smaller pores between grains while printing the core in a bed of loose sand. This is most likely the reason for the higher strength of 3D-printed cores, which is often cited in the literature. In [36], a binder content of 1.6–1.8% furan binder and 0.2% curing agent was used, resulting in a tensile strength of over 2 MPa. A review of the state-of-the-art was performed in [23], which showed that these properties are influenced by various factors such as the properties of the sand, the concentration and type of binder and activator, the printing speed (powder repainting rate), the job coordinates and print orientation of the printed sample, and the layer thickness. The study found that decreasing layer thickness and increasing binder saturation improved tensile strength, but decreased surface quality. The objective of the work presented in [37] was to achieve a molding strength of 1.8 MPa with the highest permeability and the lowest furfuryl resin content. The authors’ mathematical models showed that the properties of the molding sand were dependent on both the process parameters (such as printing speed and resolution) and the position in the printing field. The printing speed affected both strength and permeability, whereas the printing resolution determined the strength of the mold at the current settings. Moreover, it was found that bending strength was not significantly affected by the printing direction (X or Y). Reducing the printing speed increased the core density and the bending strength.

As shown in the review of the state-of-the-art, tests have been conducted to measure the strength (bending, tensile) and permeability of cores made with 3D printing technology, and these have been compared with results from the no-bake system. However, according to the authors, flexural strength may not always be the best parameter for determining the actual mechanical properties of cores. This is because it combines two parameters: compression at the top and tension at the bottom. Moreover, there is a lack of friability testing of 3D-printed cores, which is as an important technological indicator, and there has been little research into thermal deformation according to the methodology used in Central and Eastern Europe, as well as comparisons between additively manufactured cores and binders available in local markets. Furthermore, the industry has expressed an interest in knowing the achieved mechanical and technological properties of cores made using additive technology compared to conventional self-curing molding sand technology.

Therefore, the aim of this research includes the following:Determination of the mechanical and technological properties of the shell core made in additive manufacturing (Figure 1a) and comparing them with the properties of cores made with conventional technology (mixing and compaction) (Figure 1b), taking into account the size of the quartz sand grain (friability and permeability). This research aimed at identifying the possibility of reducing the consumption of resin during 3D printing in order to improve the economics of the process and reduce the formation of defects in castings of gas origin, which are closely related to the amount of organic additives introduced into the molding (core) sands.Comparison of thermal deformation of cores made by using a binder dedicated to 3D printing, with cores made by using the typical FNB system (furan no-bake system).Determining the possibility of an interchangeable use of a binder dedicated to 3D printing in typical FNB technology.Determination of binding (hardening) kinetics expressed by core strength over time, depending on the size of the grain matrix and the share of the binder in the molding (core) sand.

The strength of cores as a function of curing time (curing kinetics) was also carried out. This research aimed to compare the following:Properties of molding sands prepared with the use of two binders (conventional and dedicated to 3D printing), on the specific matrix graining (e.g., sieved fraction), using a composition dedicated to 3D printing, i.e., 1.5 parts by weight of resin and 0.4 parts by weight of hardener.The influence of the amount of binder on the properties of core sands, with a conventional binder as a reference point.

Determining the hardening time, which indicates when the core is ready to be placed in the mold cavity and has gained sufficient strength to withstand the impact of the liquid casting alloy, is crucial in terms of casting technology and the ability to produce defect-free castings. Additionally, studies have shown the impact of the hardening time (ageing) on selected core properties.

## 2. Materials and Methods

### 2.1. Materials

Table 1 lists the compositions of the molding compounds that were tested in this study.

### 2.2. Mechanical and Technological Properties of Cores (Standard Specimens—Cylindrical Shape) Made by 3D Printing Technology

Cylindrical standard samples (Ø50 × 50 mm), made by 3D printing technology from ExOne, were used for the tests. The specimens (cores) were made on quartz sand (FS001) using furfuryl resin FB001 and hardener FA001. Two types of cylindrical specimens were used:Solid core (Figure 2a)—S1_3D_ molding sand sample.Shell core filled with uncured hardener-coated sand (Figure 2b)—S1_3D_ molding sand sample.

Figure 2 shows the printed cores (cylindrical specimens) used in this study.

The following parameters were determined:Splitting tensile strength ($R_{p}^{u}$) using the LRu-2e universal tester of mechanical properties of molding sand (Multiserw-Morek, Marcyporęba, Poland) [38]; in this test, the specimen is placed horizontally between the rigid loading plates of the compression apparatus (Figure 3a). Then, the load is vertically applied and increased along the two peaks of the specimen’s diameter, top and bottom, until the specimen’s destruction (Figure 3b);

Tensile strength molding sands in a hardened state ($R_{m}^{u})$ were calculated according to the correlation given in [39], i.e., $R_{m}^{u}\;$ = 0.65$R_{p}^{u}$;Permeability (*P*^u^) was performed by the fast method on electrical apparatus type LpiR-3e [40];Friability (*S*^u^) was performed by using the HSW apparatus (Huta Stalowa Wola, Stalowa Wola, Poland). Figure 4a shows a scheme of the apparatus for determining the friability of hardened molding sand. The principle of this apparatus is as follows: A standardized cylindrical specimen (1), made of a hardened molding sand, is seated in a holder (2) by using a clamp (3). The clamped specimen is put into rotary motion at a speed of 1 rpm/s by an electric motor (4) via a gearbox (5). The electric motor is powered directly from the 220 V mains. At the top of the apparatus is placed a steel shot container (6) with a funnel-shaped bottom ending in a hole (Ø7 mm), closed by a slide bolt (7). The shot from the container falls during the marking with a tube (8) (from a height of 307 mm) on the rotating shaper and causes its abrasion (Figure 4b). The separated mass together with the shot falls into the tank (9). During the measurement, the specimen is enclosed in a shield (10), which prevents the shot and molding sand from spreading sideways.

Into the container of the apparatus is loaded 1750 g of steel shot, with a diameter of 1 mm, weighed to the nearest 1.0 g. In the case of a more abrasive molding sand, the amount of shot used in the test should be halved, i.e., 875 g should be used for testing. The specimen is seated in the holder of the apparatus, and by switching on the motor, it is made to rotate. The slide that closes the opening at the bottom of the tank is removed, and one waits until all the shot falls onto the rotating shaper. Then the motor is turned off, the sample is removed from the apparatus, and it is weighed again to the nearest 0.1 g.

Friability of hardened samples (*S^u^*), expressed as a percentage, is calculated from Equation (1):(1)$$S^{u}=\frac{Q_{1}-Q_{2}}{Q_{1}}\times 100\%$$
where:*Q*_1_—weight of the specimen before testing [g];*Q*_2_—weight of the specimen after testing [g].

The final result is the arithmetic average of the three specimens’ measurements.

### 2.3. Investigations of Mechanical and Technological Properties of Cores (Standard Samples)

As part of the study of mechanical and technological properties, splitting tensile strength, tensile strength, permeability, and friability were determined on Ø50 × 50 mm standard samples (cores), according to the methodology described in Section 2.2. The samples of molding sands S2, S3, and S4 were made by compaction using the LUZ-2e apparatus [41]. The LUZ-2e apparatus is designed for making standardized laboratory samples from loose self-curing molding and core sands, allowing 6–12 samples to be made with an equal compaction force.

In the first step, research was conducted on the composition of S1_3D_ and S2 to compare the effect of the matrix grain size on the properties of the hardened composition with the same amount of binder.

To compare the properties of the molding sands prepared with the commercial resin dedicated for 3D printing (FB001; ExOne) and the resin used in conventional technology on the same granular matrix sand with the same amount of binder, tests were conducted for molding sand S4 (conventional core preparation; FB001 resin, FA001 hardener, sieved sand fraction 0.16–0.10 mm). The final result of each selected method is the arithmetic average of a minimum of three measurements.

### 2.4. Tensile Strength as a Function of Hardening Time

The tests were carried out using standard dog-bone shaped specimens that were compacted by vibration using the LUZ-2e apparatus. The determination of the strength properties of the molding sand was carried out after 1, 3, 24, 48, and 120 h after preparation using the LRu-2e device (Figure 5). In addition to the previously mentioned molding sands’ compositions (S2, S3, and S4), the tests were supplemented with molding sand S5 (Kaltharz U404 resin 1.0 parts by weight, hardener 100T3—0.5 parts by weight, coarse sand). The final result for each test is the arithmetic average of three measurements.

### 2.5. Thermal Deformation (Hot-Distortion) Tests

A DMA apparatus (Multiserw-Morek, Marcyporęba, Poland) [42] was used to determine the hot-distortion parameters. The device was used to record the deformation of a molding or core sand samples as a function of temperature and time. The tests were performed on 3D-printed specimens and on specimens made using conventional loose self-curing molding sand (no-bake) technology. The tests were conducted on hot-distortion samples with dimensions of 114 × 25.4 × 6.3 mm, and two measurements were taken for each molding sand. To comprehensively demonstrate the influence of the resin content and the type of grain matrix, samples of the S6 molding sand were prepared with a standard amount of the binder Kaltharz U404 (1.0 part by weight), a hardener, 100T3 (0.5 parts by weight), and with a separated sand fraction of 0.16–0.10 mm (S6).

## 3. Results

### 3.1. Loss of Ignition

Determining the loss of ignition provides information on the actual amount of binder in the sample made by 3D printing technology (S1_3D_). The results of the loss of ignition (LOI) are presented in Table 2.

The determination of the loss of ignition was carried out to confirm the assumed quantitative dosing of binder components, i.e., hardener and resin. According to the manufacturer’s data, the mixture is prepared by combining quartz sand with 0.4 parts by weight of hardener (relative to sand) and 1.5 parts by weight of furfuryl resin. The loss-of-ignition test was conducted for the printed core (S1_3D_) and a sand sample with only the hardener taken from the inside of the printed shell-shaped core. By subtracting both results, the amount of resin in the core was determined.

### 3.2. Sieve Analysis

Table 3 shows the catalog data of FS001 quartz sand for 3D printing, offered by ExOne.

Table 3 shows that the average grain size of the FS001 quartz sand dedicated to 3D printing is in the range of 0.13–0.14 mm, and the sand is characterized by a high homogeneity of 89%. The main sand fraction remains sieved on sieves with a mesh size of 0.125/0.09/0.63, according to American standards.

Table 4 shows the sieve analysis of quartz sand (medium grain size; Sibelco Poland sp. z o.o., Bukowno, Poland).

In order to compare the properties of 3D-printed cores with those made using conventional self-hardening molding sands with furfuryl resin, quartz sand (Sibelco Poland sp. z o.o., Bukowno, Poland) was utilized. The main fraction of this sand was collected in sieves with a mesh size of 0.40/0.32/0.20 mm, and its average grain size was 0.39 mm (Table 3), which qualified it as “coarse” in terms of grain group. The sand was also characterized by a lower homogeneity than the FS001 sand, with a value of 58%, which is the sum of the remaining grains of the main fraction in relation to the total grains. For homogeneous sands, this sum is above 80%, whereas for non-homogeneous sands, it falls within the range of 60–80%. Quartz sands with less than 60% are classified as heterogeneous.

Table 5 shows the sieve analysis of sieved quartz sand (separated fraction; Sibelco Poland sp. z o.o., Bukowno, Poland for 3D printing.

Table 5 presents the results of the sieve analysis for separated quartz sand (Sibelco Poland sp. z o.o., Bukowno, Poland). obtained by separating the 0.16 mm and 0.1 mm grain fractions, which are the main fractions of sand for 3D printing. As a result of sieve separation, sand with a homogeneity, main fraction, grain size, and specific surface area similar to sand dedicated to 3D printing (FS001, ExOne) was obtained.

### 3.3. Mechanical Properties

#### 3.3.1. Splitting Tensile Strength

Figure 6 shows the results of the splitting tensile strength determination for standard specimens (cylindrical with dimensions of Ø50 × 50 mm) for molding sands, while Figure 7 compares the tensile strength calculated from the splitting tensile strength results, using the correlation: $R_{m}^{u}\;$ = 0.65$R_{p}^{u}$.

Figure 6 and Figure 7 demonstrate that the highest strength was achieved by 3D-printed cores, with a splitting tensile strength and tensile strength of 2.39 MPa and 1.55 MPa, respectively. The strength of the 3D-printed shell core was 1.23 MPa and 0.80 MPa, respectively. The achieved value of the shell core strength (S1_3D_(shell)) was close to the strength of the S2 molding sand. If, from a technological point of view, such a value of shell core strength is sufficient, one can consider using this solution, which, on a larger scale of production, should lead to savings in the consumption of the FB001 resin and FA001 hardener. In addition, it has been shown that the strength of cores prepared by the conventional method by compacting on sand with a sieved fraction of 0.16–0.10 mm—S3 and S4—showed significantly lower strength parameters. This may be related to the physical and chemical properties of the sand from the quartz sand (Sibelco Poland sp. z o.o., Bukowno, Poland). Moreover, different grades of binders are dedicated to different technologies, and there is no way to replace them, despite the fact that binders for 3D printing are much more expensive. The reason for this can also be attributed to the way the binder is applied to the matrix grains. In 3D printing, the resin can settle on the surface of the fine-grained sand without coating the entire surface. On the other hand, in conventional technology (mixing), a certain volume of resin introduced into the sand coats the grains over the entire surface. With a fine grain fraction, the consumption of resin increases due to the larger specific surface area. Therefore, to obtain better mechanical properties, it would be necessary to increase the amount of the binder added, which would result in higher production costs, but also an increased amount of gas that may be formed due to thermal degradation of the binder and may lead to defects in castings.

#### 3.3.2. Tensile Strength as a Function of Curing Time

Figure 8 shows the dependence of tensile strength on the curing time.

This study shows that the highest rate of setting (curing of the molding sand) can be achieved with the Kaltharz U404 resin with the 100T3 hardener on a matrix of coarse sand (S5). After 3 h, the molding sand shows a tensile strength of about 1.27 MPa, which is the highest increase in strength over time. It should also be noted that the slowest strength gain is observed for the composition S3, prepared on the same resin base but on a sieved grain fraction. Finally, after 120 h of curing, this molding sand also exhibits the lowest strength. Interestingly, the curing rate of composition S2 (typical composition of a molding sand with a furan binder in conventional technology) and composition S4 (with a binder dedicated to 3D printing) is almost identical, but only after 24 h. In the initial stage, i.e., after 3 h, the composition S4 binds much more slowly.

### 3.4. Technological Properties

#### Permeability

Figure 9 summarizes the results of the permeability measurements for the tested molding sand.

The best permeability is shown by the molding sand prepared using a matrix of coarse quartz sand with the least amount of binder (S2). The solid core (S1_3D_(solid)) and shell core (S1_3D_(shell)) made using 3D printing technology showed similar permeability, but it was the lowest among all tested systems. This result is surprising, because the inner part of the core consists only of loose sand with a hardener. In the case of molding sands prepared using separated fractions of quartz sand (Sibelco Poland sp. z o.o., Bukowno, Poland), a slightly higher permeability was achieved with molding sand S4 (a furan binder dedicated to 3D printing).

### 3.5. Friability

Figure 10 summarizes the results of the friability determination of selected samples.

An important parameter from the perspective of producing defect-free castings is the susceptibility of the molding sand to abrasion, which is related to the erosive action of the liquid casting melt jet. The figure shows that cores produced by 3D printing technology on the ExOne printer exhibit relatively low friability compared to those made by mixing and compaction through vibration. This may be related to the phenomenon described above regarding a different distribution of the set volume of the binder across the grains of the sand matrix. It should be noted that the molding sand with the lowest binder content, which is the one on a matrix of coarse sand (S2), shows the lowest friability.

### 3.6. Hot-Distortion

Figure 11 shows the dependence of the thermal deformation of the studied systems as a function of heating time.

As can be seen from Figure 11, the 3D-printed sample shows a slight susceptibility to thermal deformation, with a maximum value of 0.5%, the lowest of all the systems tested. In the initial phase of heating, the deformation is +0.30 mm, and as time passes, the direction of the deformation changes and reaches a value of 0.4 to 0.51 mm, relative to the initial value (Figure 12). After about 120 s, there is another change in the direction of deformation, followed by the destruction of the sample after about 230–240 s and a temperature of about 380 °C.

The results of the thermal deformation tests on molding sands made on the same grain matrix (quartz sand Sibelco Poland sp. z o.o., Bukowno, Poland separated to an average grain size of 0.13 mm, corresponding to the average grain size of FS001 sand for 3D printing), with the same share of binder components (1.5 parts by weight of resin and 0.4 parts by weight of hardener), but with different binders, i.e., a typical furan binder (S3 molding sand: Kaltharz U404 resin, 100T3 hardener), and a binder dedicated to 3D printing technology (S4 molding sand: FB001 resin, FA001 hardener) showed significant differences in strain size that may be important from a technological point of view (Figure 12a,b). The deformation in the case of the S3 molding sand was between 2.75–3.00%, while the S4 molding sand was larger, ranging from 3.30–3.50%. Therefore, it can be concluded that the level of deformation is not only due to the amount of the binder in the sand, but also to the method of preparing the molding (core) sand, as evidenced by the low deformation of the printed sample, which was only 0.5%.

Figure 12c,d compare the thermal deformation of molding sands prepared using the Kaltharz U404 resin with the 100T3 hardener with varying amounts of binder components (1.0 and 0.5 parts by weight) in relation to the grain matrix. These amounts correspond to the commonly used shares of the binder in conventional self-hardening sands with furfuryl resin. The grain matrix was composed of quartz sand from Sibelco Poland sp. z o.o., Bukowno, Poland, either unseparated or coarse (Table 4), or separated to a grain size equivalent to the sand used for 3D printing (Table 5).

As shown in Figure 12c,d, the size of the grain of the sand matrix has little effect on the size of thermal deformation, and any differences may result from a greater number of impurities or very fine fractions of silica in the S6 molding sand. The course of the curves showing the thermal deformation is similar, and they are characterized by deformation with an addition in the initial heating phase (about 25 s from the start of the measurement), then it changes sign in the negative direction and reaches the maximum deformation after about 120 s. In both cases, for the second tested sample, another change in the strain direction was noted, with a clearly marked maximum, and after about 200 s, both samples were destroyed.

Figure 12a,d show a comparison of the thermal deformation of the S3 and S6 sands—made on the basis of the same matrix, the same binder, with a lower content of binder used in the S6 sand.

Figure 12a,d demonstrates that reducing the content of furfuryl resin in the molding sand by 0.5 parts by weight results in a more than two-fold reduction in thermal deformation.

## 4. Conclusions

Based on this study, the following conclusions can be made:The highest mechanical strength is achieved by cores made by using 3D printing technology;Shell (3D-printed) cores are characterized by lower mechanical strength but are similar to cores made with conventional technology. Considering the high cost of purchasing sand, resin, and hardener, one can consider switching to such a production system, taking into account the proper thickness of the core wall;The use of a binder dedicated to 3D printing in conventional technology (mixing and compacting) is not justified both from a technological and economic point of view;Printed cores (solid and shell) have similar a permeability, but are the lowest of all the tested systems;The friability of 3D-printed cores is at a low level, due to their high mechanical properties;Three-dimensionally printed cores show a negligible susceptibility to thermal deformation and a resistance to high temperature;The most significant influence on the thermal deformation of molding sands is the binder content. It has been noted that molding sands with a binder dedicated to 3D printing (FB001 and FA001) show greater susceptibility to deformation compared to typical binders (Kaltharz U404 + 100T3);By reducing the amount of binder by 0.5 parts by weight, the deformation of the sample is reduceable by up to two times.

The obtained results show that it is possible to reduce the amount of binder in the entire volume of the core by using a shell core, while maintaining sufficient mechanical strength from the point of view of casting-production technology and reducing the susceptibility of the core to thermal deformation. The cores also maintain good technological properties, i.e., friability and gas permeability, and the thermal deformation of 3D-printed cores is lower than those made with conventional technology. In addition, the reduction of the amount of the binder positively affects the size of the core deformation when in contact with high temperature.

## Figures and Tables

**Figure 1 materials-16-03339-f001:**
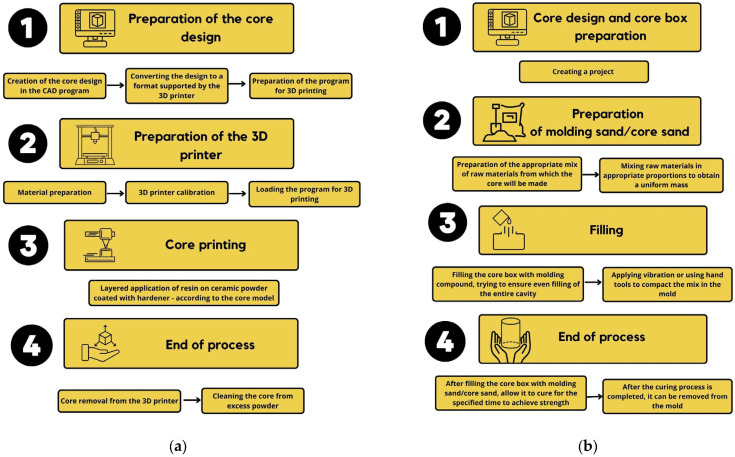
Schematic process of (**a**)—the additive manufacturing of sand cores (3D printing), (**b**)—the conventional manufacturing of sand cores.

**Figure 2 materials-16-03339-f002:**
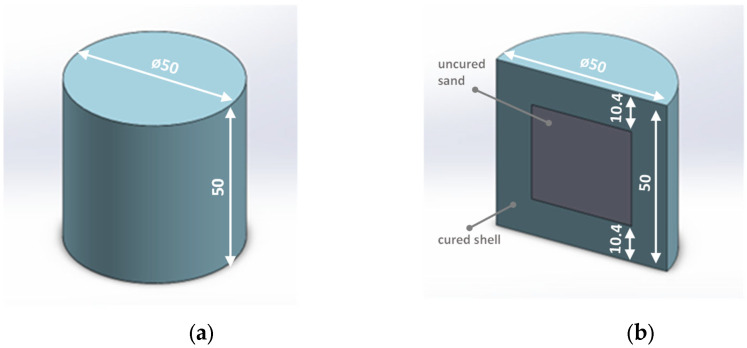
3D-printed cores (cylindrical samples): (**a**)—solid, hardened throughout the sample volume; (**b**)—shell-filled with uncured sand with a hardener (cross-sectional view).

**Figure 3 materials-16-03339-f003:**
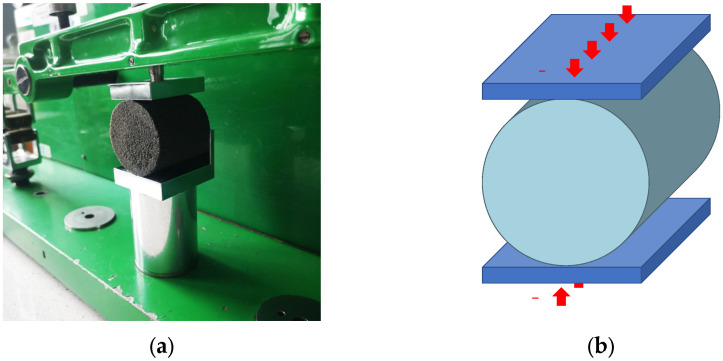
The position of the standard cylindrical specimen in the apparatus (**a**), and the direction of the applied splitting force (**b**).

**Figure 4 materials-16-03339-f004:**
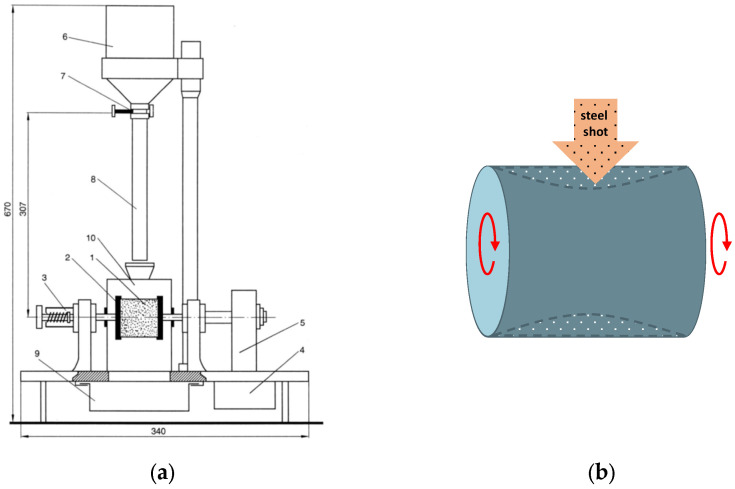
Scheme of apparatus for determining the friability of the production of the HSW (Huta Stalowa Wola, Stalowa Wola, Poland) (**a**), description in the text; the direction of the steel shot falling on the surface of the rotating specimen (**b**).

**Figure 5 materials-16-03339-f005:**
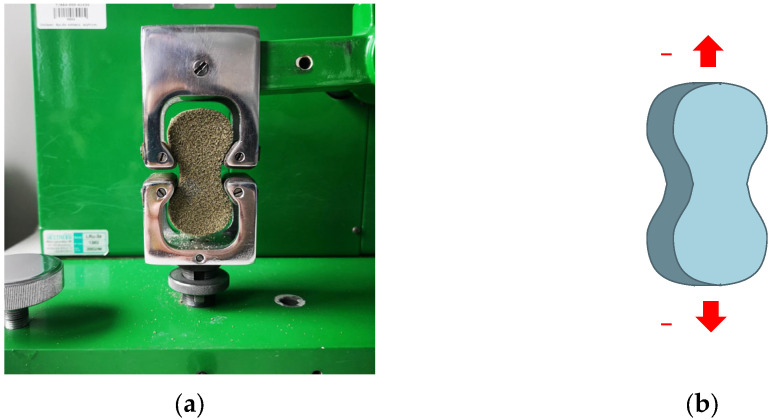
The position of the standard dog-bone shaped specimen in the apparatus (**a**), and the direction of the applied tensile force (**b**).

**Figure 6 materials-16-03339-f006:**
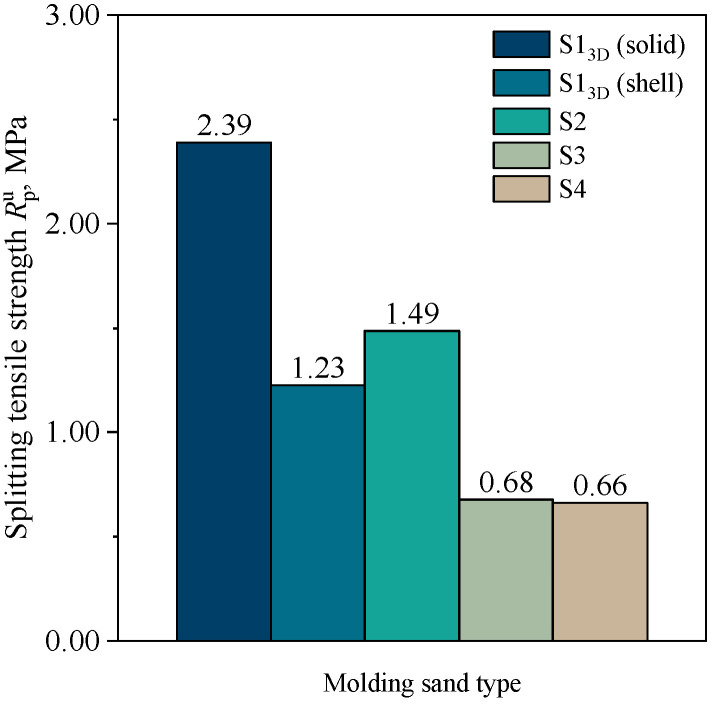
Splitting tensile strength of molding sands: S1_3D_(solid), S1_3D_(shell), S2, S3, and S4.

**Figure 7 materials-16-03339-f007:**
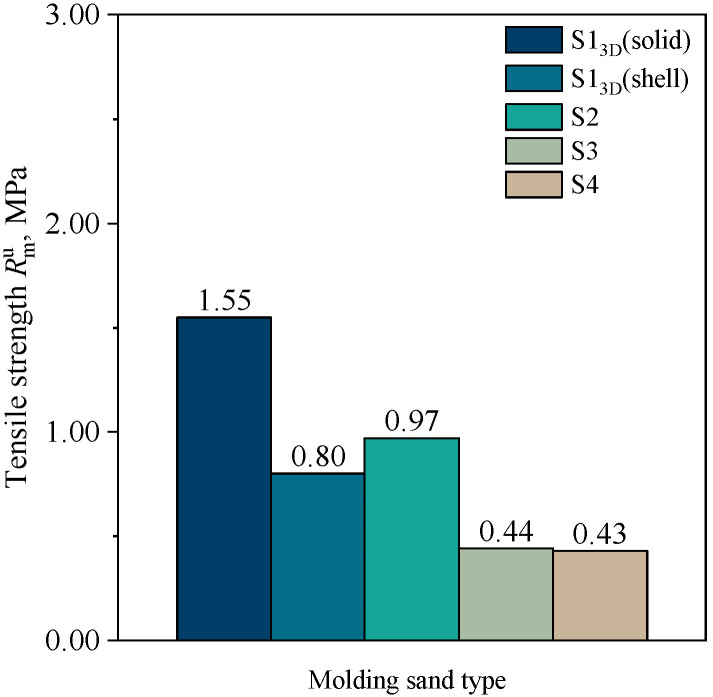
Tensile strength of molding sands: S1_3D_(solid), S1_3D_(shell), S2, S3, and S4.

**Figure 8 materials-16-03339-f008:**
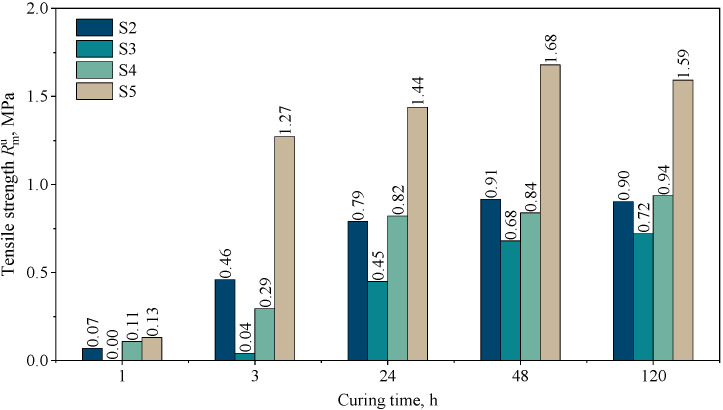
Tensile strength of molding sands: S2, S3, S4, and S5 as a function of curing time.

**Figure 9 materials-16-03339-f009:**
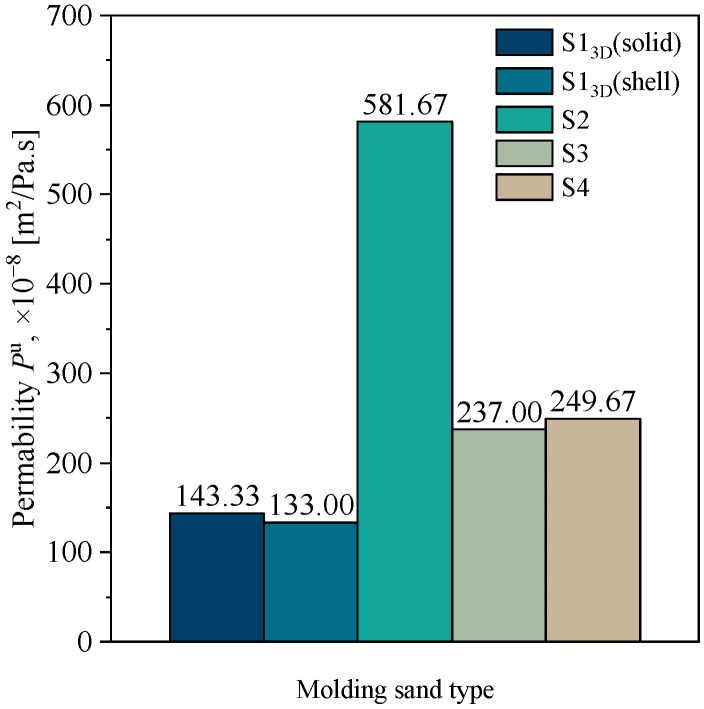
Permeability of molding sands: S1_3D_(solid), S1_3D_(shell), S2, S3, and S4.

**Figure 10 materials-16-03339-f010:**
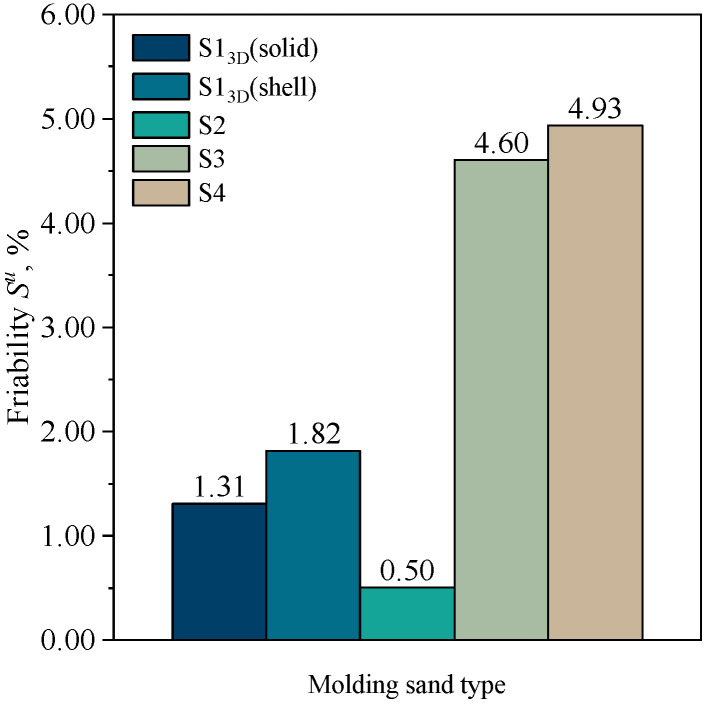
Friability of molding sands: S1_3D_(solid), S1_3D_(shell), S2, S3, and S4.

**Figure 11 materials-16-03339-f011:**
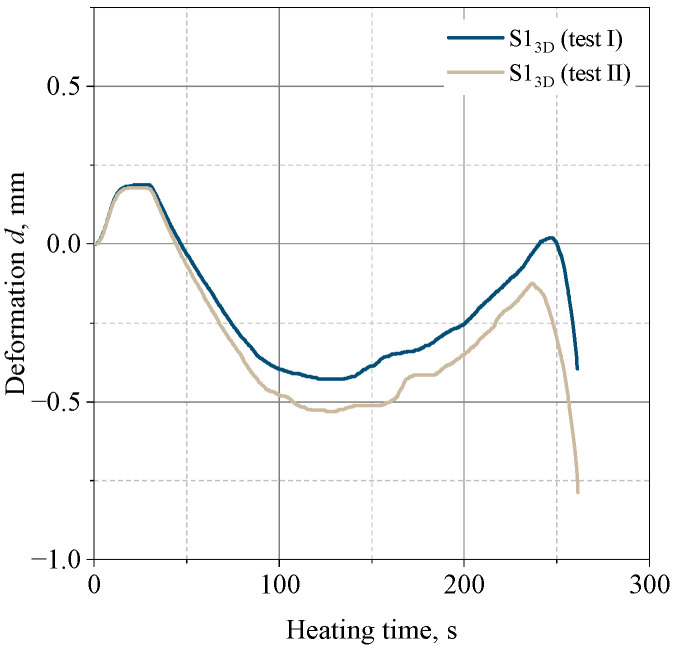
Thermal deformation of molding sand S13D—shapes made by a 3D printing technique (S13D composition).

**Figure 12 materials-16-03339-f012:**
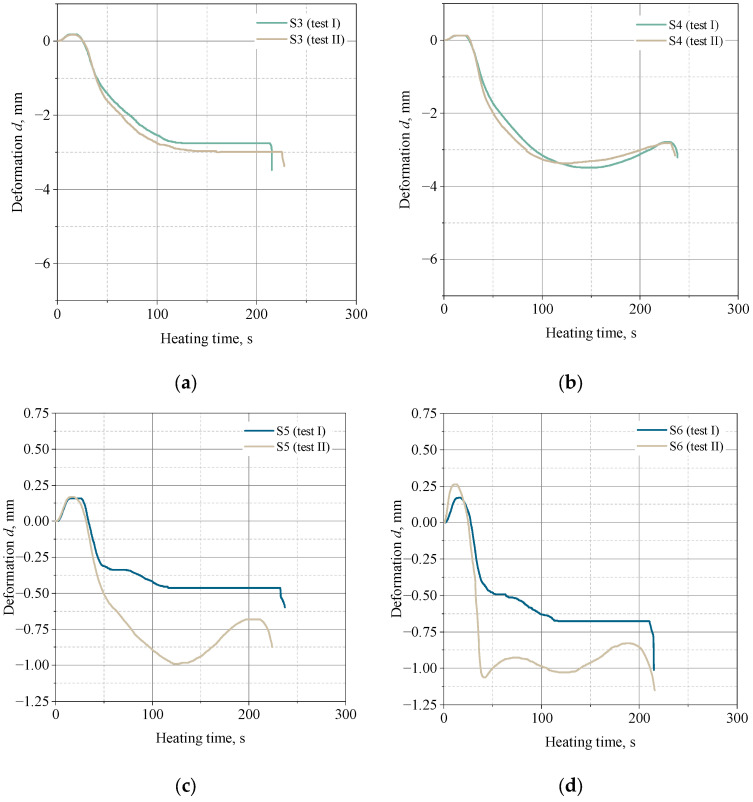
Thermal deformation of the molding sand: (**a**)—S3 (conventional furan resin Kaltharz U404 + 100T3, separated sand fraction 0.16–0.10 mm); (**b**)—S4 (binder FB001 dedicated to 3D printing, separated sand fraction 0.16–0.10 mm); (**c**)—S5 (conventional composition Kaltharz U404 + 100T3, coarse sand), (**d**)—S6 (conventional composition Kaltharz U404+100T3, separated sand fraction 0.16–0.10 mm).

**Table 1 materials-16-03339-t001:** Summary of molding sand systems tested for selected properties.

Molding Sand	Core Preparation Method	Binder, Part by Weight		Hardener, Part by Weight		Sand Grain Matrix, Part by Weight		
		FB001 ^a^	Kaltharz U404 ^b^	FA001 ^a^	Aktivator 100T3 ^b^	FS001 ^a^	Coarse Sand ^c^ (Main Fraction 0.40/0.32/0.20)	Sieved Sand ^c^ (Fraction 0.16/0.10)
S1_3D_	3D printing	~1.5	–	~0.4	–	~100	–	–
S2	conventional: mixing/compacting	–	1.5	–	0.4	–	100	–
S3	conventional: mixing/compacting	–	1.5	–	0.4	–	–	100
S4	conventional: mixing/compacting	1.5	–	0.4	-	–	–	100
S5	conventional: mixing/compacting	–	1.0	–	0.5	–	100	–
S6	conventional: mixing/compacting	–	1.0	–	0.5	–	–	100

^a^ ExOne, dedicated to 3D-printed cores, ^b^ Huttenes-Albertus, dedicated to conventional prepared cores, ^c^ Sibelco Poland sp. z o.o., Bukowno, Poland; quartz sand.

**Table 2 materials-16-03339-t002:** Results of loss-of-ignition measurement.

Molding Sand Type	Loss of Ignition (Average), %
S1_3D_ (solid sample: FB001 + FA001 + FS001)	1.88
S1_3D_—loose sand from shell specimen (sand FS001 + hardener FA001)	0.39
Difference (resin FB001 content in sample)	1.49

**Table 3 materials-16-03339-t003:** Selected data of FS001 quartz sand for 3D printing, offered by ExOne.

Average grain size	0.13–0.14 mm
Theoretical specific surface area	17.6 m^2^/kg
Uniformity ratio	89%
AFS number	97
Sieve analysis, %	
>0.71	0.0
>0.55	0.0
>0.355	0.0
>0.25	0.0
>0.18	3.5
>0.125	60.7
>0.09	29.1
>0.063	6.0
<0.063	0.7

**Table 4 materials-16-03339-t004:** Sieve analysis of quartz sand (Sibelco Poland sp. z o.o., Bukowno, Poland).

Main fraction	0.40/0.32/0.20
Medium grain size	0.39 mm
Theoretical specific surface area	6.5 m^2^/kg
Uniformity ratio	58%
AFS number	40.24
Sieve analysis, %	
1.600	0.00
0.800	0.48
0.630	6.13
0.400	40.09
0.320	20.44
0.200	25.99
0.160	4.54
0.100	2.28
0.071	0.04
0.056	0.00
button	0.01

**Table 5 materials-16-03339-t005:** Selected data of sieved quartz sand (separated fraction; Sibelco Poland sp. z o.o., Bukowno, Poland) for 3D printing.

Main fraction	0.16/0.10/0.071
Medium grain size	0.13 mm
Theoretical specific surface area	17.8 m2/kg
Uniformity ratio	87%
AFS number	10.47
Sieve analysis, %	
1.600	0.00
0.800	0.00
0.630	0.00
0.400	0.00
0.320	0.00
0.200	2.42
0.160	12.00
0.100	74.78
0.071	8.92
0.056	0.90
button	0.68

## Data Availability

The data is contained within the article and/or is available upon request from the corresponding author.

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
