# Peer review of "3D Printed (Binder Jetting) Furan Molding and Core Sands—Thermal Deformation, Mechanical and Technological Properties"

_materials, 2023, doi:10.3390/ma16093339_

Round 1

Reviewer 1 Report

Please see comments and suggestions as below:

1. For table 1, the author should state the rationale that why these combinations were selected. What can be the main take away from this systematic study.  

2.In results section (for most tests including LOI, sieve analysis, permeability, and hot distortion), the author needs to provide more comprehensive analysis/explanation for results instead of only presenting results. 

3. Could the author explain why with fine-grained sand, resin can settle on the surface of grains without coating the surface. 

Author Response

Dear reviewer,
the response to the review is in the attachment.

Reviewer 2 Report

This article compares the properties of furan resin molding and core sands made using 3D printing technology with traditional molding sands. Two types of resins were used, and the effect of grain matrix fraction size on the properties of the molding sands was studied. Thermal deformation tests were conducted, and the mechanical and technological properties of 3D printed standard shapes were determined for solid and shell core designs. The results were compared to those produced by traditional free-flowing self-hardening molds prepared on two types of grain matrix.

The study found that 3D printed cores have the highest mechanical strength, while shell cores have lower strength but are still comparable to traditional technology. Using a 3D printing binder in traditional technology is not justified. 3D printed cores have low permeability and friability due to high mechanical properties, and negligible susceptibility to thermal deformation and high temperatures. The binder content has the most significant influence on the thermal deformation of molding sands, with sands using a 3D printing binder being more susceptible to deformation than traditional binder. By reducing the binder content by 0.5 parts by weight, deformation can be reduced by up to 2 times.

While the comparative data is instructive for researchers in both academia and industry, the novelty of the study and the reason for material selection are not clear. In addition, I suggest adding a schematic diagram of the manufacturing process with both 3D printing and traditional methods, some take-home messages in the Conclusion section, and shortening the Introduction. These revisions will enhance the clarity and impact of the manuscript.

Author Response

(The authors gave the same response as above.)

Reviewer 3 Report

The authors studied the impact of a wide range of parameters (fabrication method, resin type, grain type matrix) on the thermo-mechanical properties of core sands (splitting tensile strength, tensile strength, permeability, friability, and thermal deformation). The subject of the study is of interest, but the manuscript lacks clarity. Improving the writing and the presentation of results could significantly improve the readability of the manuscript.

Specific comments:

1.      Abstract.

The abstract should be extensively revised. The English writing must be improved, a general sentence introducing the subject should be included, and the objectives of the study along with the different parameters studied could be mentioned in a much more clear way than it is currently.

2.      Introduction.

The introduction could be significantly improved. It is difficult for the reader to understand the subject and the state of the art of this field. A first paragraph introducing the subject by giving a general overview of what are core sands and why they are important could help the reader as well. There are few repetitions and the objectives of the study are confused and not clearly stated. For example:

2.1. Line 131: “an additional objective”.

What is the first objective? The paragraph preceding this sentence is not very clear…

2.2. Line 142: What is “eight-shaped”? What does that mean?

2.3. Line 144: It is weird to indicate S3 and S4 at this moment. It would make more sense to present first the entire range of parameters studied in the introduction without mentioning the name of samples. Next, in Section 2, all samples along with their labels can be introduced together in numerical order.

2.4. Line 148: Same issue here with S1 and S5. At this stage, the reader wonders why there is no S2.

2.5. Figure 1: Please indicate the dimensions directly on the drawings and in the caption (in particular the diameter and height, in addition to the wall thickness already indicated).

3.      Section 2.

3.1. Table 1: It would be interesting to add a column indicating the method of fabrication. It is also weird to have first S2b followed by S2a.

3.2. Section 2.1., Line 175.

A drawing would greatly help to understand the test. It could be added in Figure 2. Besides, the test of friability is very much detailed, but I am not sure if it helps the understanding of the manuscript and I wonder if this information could not be moved in a supplementary file (optionally).

4.      Section 3.

4.1. Section 3.1. There is no comment on the result of loss of ignition. What is the conclusion to draw from these results?

4.2. Section 3.2. Here also, there is no comment on the results. It would be appreciable to have a couple of lines summarizing the main properties of each sand. Also, in Line 265 and 268, the number of the table is not indicated. The reader does not know which table the authors refer to.

4.3. Section 3.3. While the results of Figure 3 are commented, the results of Figure 4 are not. If they are not useful for the characterization, the authors could remove it (optionally movable to a supplementary file). Alternatively, the authors could merge Figure 3 and 4 into Figure 3a and 3b, and give a brief comment on the results of Figure 3b.

4.4. Figure 5. The mention “as a function of time” should be added in the caption of Figure 5 (Line 311).

4.5. Figure 6. Samples S2a and S2b are presented in the Figure but are not indicated in the caption. Please address this. In addition, the paragraph starting Line 330 explaining the results of Figure 6 was barely understandable grammatically. Please review this paragraph.

4.6. Section 3.5. Please remind the reader one more time which sample number correspond to 3D printing technology and cores made by mixing and compaction, by indicating in brackets the corresponding number (Line 345-346).

4.7. Section 3.6. Line 351. It should be noted “The Figures 8 to 12”. The authors should review how to present this section. For example, Line 357, it is mentioned “the lowest of all the systems tested”, but the Figures come only later in the text. Line 374: Do the authors mean “The Figures 9 to 12”?

5.      References:

References [15], [20], [22], and [23] are not well formatted or have errors. Please review the information provided and double check the rest of the list.

Author Response

(The authors gave the same response as above.)

Round 2

Reviewer 2 Report

My concerns have been well addressed by the authors. I am happy to recommend its publication in this journal.